# Resveratrol Promotes Mitochondrial Biogenesis and Protects against Seizure-Induced Neuronal Cell Damage in the Hippocampus Following Status Epilepticus by Activation of the PGC-1α Signaling Pathway

**DOI:** 10.3390/ijms20040998

**Published:** 2019-02-25

**Authors:** Yao-Chung Chuang, Shang-Der Chen, Chung-Yao Hsu, Shu-Fang Chen, Nai-Ching Chen, Shuo-Bin Jou

**Affiliations:** 1Department of Neurology, Kaohsiung Chang Gung Memorial Hospital, Kaohsiung City 83301, Taiwan; ycchuang@cgmh.org.tw (Y.-C.C.); chensd@adm.cgmh.org.tw (S.-D.C.); fangoe1@yahoo.com.tw (S.-F.C.); naiging@yahoo.com.tw (N.-C.C.); 2Institute for Translation Research in Biomedicine; Kaohsiung Chang Gung Memorial Hospital, Kaohsiung City 83301, Taiwan; 3College of Medicine, Chang Gung University, Taoyuan City 33302, Taiwan; 4Department of Neurology, Faculty of Medicine, College of Medicine, Kaohsiung Medical University, Kaohsiung City 80708, Taiwan; cyhsu61@gmail.com; 5Department of Biological Science, National Sun Yat-sen University, Kaohsiung City 80424, Taiwan; 6Department of Neurology, Mackay Memorial Hospital and Mackay Medical College, Taipei 252, Taiwan

**Keywords:** resveratrol, PGC-1α, mitochondrial biogenesis, status epilepticus, hippocampus

## Abstract

Peroxisome proliferator-activated receptor gamma coactivator 1-alpha (PGC-1α) is known to regulate mitochondrial biogenesis. Resveratrol is present in a variety of plants, including the skin of grapes, blueberries, raspberries, mulberries, and peanuts. It has been shown to offer protective effects against a number of cardiovascular and neurodegenerative diseases, stroke, and epilepsy. This study examined the neuroprotective effect of resveratrol on mitochondrial biogenesis in the hippocampus following experimental status epilepticus. Kainic acid was microinjected into left hippocampal CA3 in Sprague Dawley rats to induce bilateral prolonged seizure activity. PGC-1α expression and related mitochondrial biogenesis were investigated. Amounts of nuclear respiratory factor 1 (NRF1), mitochondrial transcription factor A (Tfam), cytochrome c oxidase 1 (COX1), and mitochondrial DNA (mtDNA) were measured to evaluate the extent of mitochondrial biogenesis. Increased PGC-1α and mitochondrial biogenesis machinery after prolonged seizure were found in CA3. Resveratrol increased expression of PGC-1α, NRF1, and Tfam, NRF1 binding activity, COX1 level, and mtDNA amount. In addition, resveratrol reduced activated caspase-3 activity and attenuated neuronal cell damage in the hippocampus following status epilepticus. These results suggest that resveratrol plays a pivotal role in the mitochondrial biogenesis machinery that may provide a protective mechanism counteracting seizure-induced neuronal damage by activation of the PGC-1α signaling pathway.

## 1. Introduction

Status epilepticus, or the condition of continuous epileptic seizures, is a major neurological and medical emergency that is associated with significant morbidity and mortality [1]. In studies of both humans and animal models, results showed many changes and cascades of events at the cellular level, including activation of glutamate transmission, changes in the ingredient of γ-aminobutyric acid and glutamate receptors, activation of inflammatory cytokines, increased oxidative stress, mitochondrial dysfunction, activations of neuronal plasticity, and activation of late cell death pathways, which may play crucial roles in the development of brain damage and the decline of cognition [1,2,3,4]. 

Mitochondria are essential organelles in cells that participate in energy exchange, regulation of signaling cascades, oxidant formation, and the life and death of cells [2,3,4]. We have previously demonstrated that experiment-induced prolonged seizures cause dysfunction of enzyme activity in complex I of the mitochondrial respiratory chain and damage to mitochondrial ultrastructure in the hippocampus [5,6,7]. Our subsequent studies revealed that prolonged seizures lead to a decline in mitochondrial complex I enzyme activity, which raises oxidative and nitrosative stress, increases cytochrome *c* release from the mitochondria to the cytosol, and triggers the activation of caspase, leading to apoptotic cascade and causing cell death in the hippocampus [6,7,8]. In recent years, mitochondrial dynamics has been acknowledged as a crucial process affecting cell death and survival; in particular, mitochondrial fission happens as an early event in the apoptotic process and results in neuronal cell death in various cerebral insults [9,10]. Several studies, including ours, showed that seizure-affected mitochondrial fission expression with neuronal damage and alteration of mitochondrial dynamic protein expression can provide a protective effect opposing seizure-induced hippocampal neuronal damage [5,11,12].

Polyphenols belong to a category of chemicals that naturally occur in plants, including flavonoids and nonflavonoids [13]. Recently, many human intervention trials and animal studies have provided evidence for protective effects of various (poly)phenol-rich foods against various chronic diseases. Resveratrol (3,5,4′-trihydroxy-*trans*-stilbene), primarily found in red grapes/wine, is a nonflavonoid polyphenol [13]. It is present in a variety of plants, including the skin of grapes, blueberries, raspberries, mulberries, and peanuts [13,14]. Resveratrol has been reported to improve survival rates, endothelium-dependent smooth muscle relaxation, cardiac contractility, and mitochondrial function in a hypertensive model of heart failure. Meta-analysis studies indicated that dietary resveratrol and flavonoids are associated with decreased risk of all-cause mortality and mortality of cardiovascular diseases [13,15,16]. It has also been shown to offer protective effects against a number of neurodegenerative diseases, such as Parkinson’s disease, aging and depression, and many cancers [14,17,18,19]. 

Whereas resveratrol exerts a wide range of beneficial effects on many diseases, its mechanisms have not yet been clearly elucidated. A number of literatures have revealed that resveratrol is provided with anti-inflammatory, antioxidative and metal-chelating properties [14,17,20]. Besides its antioxidant and anti-inflammatory properties, growing evidence showed that resveratrol can activate sirtuin 1 (SIRT1), a class III lysine-deacetylase, which plays an important role in the protective mechanism of resveratrol [20]. Several significant features of SIRT1 were demonstrated in brain neuronal cells [21]. These include the ability to preserve mitochondria function and modulate responses to DNA damage [22] and functionally interact with peroxisome proliferator-activated receptor gamma coactivator 1-alpha (PGC-1α), and it may have an important role in mitochondrial biogenesis [23]. A recent study showed that PGC-1α activation augments the mitochondrial antioxidant signaling pathway in status epilepticus [24]. PGC-1α was also previously reported to improve the severely impaired ability of mitochondrial biogenesis in the hippocampus in rats with chronic seizures [5,25].

As mitochondrial biogenesis is an important feature of the PGC-1α pathway, it is tempting to postulate that this pathway could be affected during status epilepticus and may confer protective effects against seizure-induced neuronal damage through the change of mitochondrial biogenesis machinery expression in the hippocampus. Resveratrol may contribute to PGC-1α–related mitochondrial biogenesis and further protect against hippocampal neuronal cell death following status epilepticus. We have validated this hypothesis using an experimental status epilepticus model in the present study.

## 2. Results

### 2.1. Temporal Changes of PGC-1α Expression in the Hippocampal CA3 Subfield Following Experimental Status Epilepticus

Our first series of experiments examined whether PGC-1α expression in the hippocampal CA3 subfield exhibited changes following experimental status epilepticus. After unilateral microinjection of kainic acid (KA) into the left CA3 region, real-time PCR analysis (Figure 1A) revealed that *pgc-1α* mRNA have a significant increase in the right hippocampal CA3 subfield 1 h after the induction of experimental status epilepticus, followed by a significant reduction that returned to baseline at 24 h. In addition, Western blot analysis showed a significant increase of PGC-1α protein levels in total proteins extracted from the right hippocampal CA3 subfield 1–24 h after the induction of experimental status epilepticus that peaked at 6 h (Figure 1B).

### 2.2. Temporal Changes of Mitochondrial Biogenesis Machinery Expression in the Hippocampal CA3 Subfield Following Experimental Status Epilepticus

To demonstrate the temporal change of mitochondrial biogenesis machinery expression following experimental status epilepticus, we first showed nuclear respiratory factor 1 (NRF1) expression in total protein prepared from the right hippocampal CA3 subfield, which revealed a significant increase of expression of NRF1 from 3 to 24 h, with peak level at 6 h after KA treatment (Figure 2A). We further extracted nuclear proteins from the hippocampal CA3 subfield to show the authentic activity of NRF1 as a transcription factor and revealed increasing DNA binding activity from 1–6 h after KA treatment (Figure 2B). 

We therefore used mitochondrial protein fraction to perform western blot analysis, which showed temporal change of mitochondrial transcription factor A (Tfam) expression and revealed increased expression in 3–24 h under KA treatment (Figure 2C). We further showed the extent of mitochondrial biogenesis and determined whether it was compatible with NRF1/Tfam expression. Western blot analysis revealed a significant increase of the mitochondrial DNA-encoded polypeptide cytochrome *c* oxidase 1 (COX1) in the hippocampal CA3 subfield 6–24 h after KA treatment (Figure 2D). We also used the long PCR method to quantify mitochondrial DNA, which can be used as an index of mitochondrial DNA (mtDNA) with sufficient integrity [26,27]. It revealed increased mtDNA content in 3–24 h in the right hippocampal CA3 subfield after KA treatment (Figure 2E). 

### 2.3. Effect of Resveratrol on PGC-1α Expression in the Hippocampus Following Experimental Status Epilepticus

In order to determine the causal effect of resveratrol on PGC-1α in this experimental paradigm, we further employed western blot analysis to test the resveratrol on PGC-1α expression in the hippocampus following status epilepticus. Bilateral microinjection of the PGC-1α activator, resveratrol (100 µmol), into the hippocampal CA3 region significantly increased the expression of PGC-1α in the CA3 subfield 6 h after the elicitation of sustained hippocampal seizure discharges (Figure 3A). To confirm the augmented activation of PGC-1α by resveratrol treatment demonstrated in our biochemical analysis, we examined the intracellular expression of PGC-1α in hippocampal CA3 neurons using double immunofluorescence staining (Figure 3B). Compared to sham-control (Figure 3Ba–c), there was an increase in PGC-1α immunoreactivity in neuronal cells in the right CA3 area 6 h after KA-induced status epilepticus (Figure 3Bd–f). Moreover, pretreatment of resveratrol (100 µmol) augmented PGC-1α immunoreactivity in the hippocampal CA3 neurons (Figure 3Bg–i).

### 2.4. Effect of Resveratrol on Mitochondrial Biogenesis Machinery Expression in the Hippocampal CA3 Subfield Following Experimental Status Epilepticus

We further employed western blot analysis to test the effects of resveratrol on mitochondrial biogenesis machinery expression in the hippocampal CA3 subfield following status epilepticus. After pretreated microinjection with resveratrol (100 µmol) into the bilateral CA3 subfield before KA-induced status epilepticus, western blot analysis revealed an increase of NRF1 protein level in hippocampal CA3 neurons 6 h after KA treatment compared with sham animals and animals with KA-induced status epilepticus without resveratrol treatment (Figure 4A). With resveratrol (100 µmol) treatment, DNA binding activity of NRF1 also increased in hippocampal CA3 neurons 6 h after KA treatment (Figure 4B). Therefore, we demonstrated that pretreated microinjection with resveratrol (100 µmol) enhanced Tfam expression 24 h after KA treatment compared with sham animals and animals with KA-induced status epilepticus (Figure 4C). After pretreated microinjection with resveratrol (100 µmol) into the bilateral CA3 subfield, the effects of mitochondrial biogenesis in the hippocampus increased the content of both mitochondrial protein COX1 (Figure 4D) and mtDNA (Figure 4E). These results may reveal the potential causal relationship between resveratrol-PGC-1α signaling and mitochondrial biogenesis.

### 2.5. Effect of Resveratrol on Apoptosis and Neuronal Survival in the Hippocampal CA3 Subfield Following Experimental Status Epilepticus

To further elucidate the neuroprotective effect of resveratrol-PGC-1α signaling on hippocampal damage, we investigated the effects of resveratrol on KA-induced hippocampal neuronal cell death. Pretreatment with resveratrol (100 µmol) attenuated the extent of caspase-3 expression in the hippocampal CA3 subfield 7 days after KA-induced status epilepticus (Figure 5A). A decreased extent of neuronal damage with resveratrol (100 µmol) treatment was demonstrated in the hippocampal CA3 subfield in both qualitative (Figure 5B) and quantitative (Figure 5C) analysis of DNA fragmentation, an index for apoptosis, 7 days after the induction of status epilepticus. These results may imply that resveratrol-PGC-1α signaling involves neuronal survival following status epilepticus. Decreased neuronal damage in the hippocampus with pretreatment of resveratrol (100 µmol) was also shown by immunofluorescent staining (Figure 5D) using terminal deoxynucleotidyl transferase dUTP nick end labeling (TUNEL) immunoreactivity in hippocampal CA3b neurons on the right side with co-immunofluorescence staining with 4′,6-diamidino-2-phenylindole (DAPI).

## 3. Discussion

With a clinically related animal model of status epilepticus, the present study shows changes of mitochondrial biogenesis machinery induced by sustained epileptic seizures that upregulated PGC-1α expression, and resveratrol, the activator of PGC-1α, was accompanied by increased PGC-1α expression and promotion of mitochondrial biogenesis. Furthermore, resveratrol reduced activated caspase-3 activity and attenuated neuronal cell damage in the hippocampus. These results may indicate that resveratrol activates the PGC-1α pathway, which involves the mitochondrial biogenesis machinery and exerts an endogenous protective mechanism in the hippocampus following status epilepticus.

It is well recognized that the sirtuin family, with deacetylation reaction, plays a critical role in many physiological activities, such as regulation of transcription, DNA damage repair, protein secretion, and metabolic action. These proteins hold potential as therapeutic targets for various human disorders, including malignancies, metabolic disorders, and neurodegenerative diseases [14,28]. SIRT1, the most-studied sirtuin family, was initially shown to deacetylate histones, but was later also shown to deacetylate several other protein targets, including PGC-1α, FOXO, P53, Notch, HIF1α, and others [22,29]. SIRT1 is distributed in various adult brain areas, with higher expression in the cortex, cerebellum, hypothalamus, and hippocampus and lower expression in white matter [30]. It has been reported that SIRT1 is mainly expressed in neurons [30,31]. Previous studies showed the neuroprotective properties of SIRT1 in both acute and chronic neurological diseases with various mechanisms [32,33,34]. Recent studies show the ability of resveratrol to exert neuroprotective effects through activation of the SIRT1-PGC-1α signaling pathway, decreasing oxidative stress and promoting mitochondrial biogenesis in the neuronal cells, decreasing neuronal and glial cell inflammation, and involving the pathway of neuron cell death and survival [14,17,20,28]. Many studies have demonstrated the ability of resveratrol to exert neuroprotective effects in neurodegenerative diseases [14], such as Alzheimer’s disease, Parkinson’s disease, Huntington’s disease, acute stroke [35,36], and epilepsy and status epilepticus [37,38,39,40]. 

In our earlier work using the KA-induced experimental status epilepticus model, we showed that prolonged seizure caused dysfunction of complex I respiratory chain enzyme activity and mitochondrial ultrastructure damage in the hippocampus [6]. We further investigated mitochondrial dysfunction induced by nitrosative stress. We found that prolonged seizures prompted nitric oxide and superoxide anion production and peroxynitrite formation, and compromised mitochondrial respiratory enzyme activity, causing cytochrome *c*/caspase-3–dependent apoptotic cell death in the hippocampal CA3 subfield [7,8]. The mitochondrial uncoupling protein 2 (UCP2) is known as an endogenous neuroprotective molecule in many neurological disorders [41,42]. With this status epilepticus model, we demonstrated that activation of peroxisome proliferator-activated receptor γ (PPARγ) increased mitochondrial UCP2 expression, decreased mitochondrial translocation of Bax, reduced cytosolic release of cytochrome *c* by stabilizing the mitochondrial transmembrane potential, and lessened apoptotic neuronal cell death in the hippocampus [43]. Recently, we also investigated the role of dynamin-related protein 1 (Drp1), the major mitochondrial fission protein, in the hippocampus following status epilepticus. We showed that activation of phosphorylation of Drp1 at serine 616 (p-Drp1(Ser616)), related to seizure-induced neuronal damage and lessened p-Drp1(Ser616) expression, can reduce mitochondrial fission and decrease mitochondrial dysfunction and oxidation, offering a protective strategy against seizure-induced hippocampal neuronal damage [5]. Our recent studies indicated that the PGC-1α–related pathway may enhance mitochondrial proteins UCP2 and superoxide dismutase 2 to counteract excessive reactive oxygen species (ROS) and increase mitochondrial biogenesis in the hippocampus after cerebral ischemia [44,45,46] and experimental status epilepticus [5,43]. It was suggested that enhancing the ability of mitochondrial biogenesis may be a protective strategy in various neurological diseases [47,48,49,50], including chronic epilepsy and status epilepticus [5,37,38,39,40,43]. 

The mitochondrial biogenesis machinery represents a complex biological process that controls the biogenesis of mitochondria and the maintenance of mtDNA. Most respiratory proteins and all proteins and enzymes that are involved in mtDNA replication, transcription, and translation as well as gene products necessary for the numerous mitochondrial functions stem from nuclear genes [50,51]. Nuclear respiratory factors NRF1 and NRF2, functioning as transcriptional regulators, are essential subunits of the oxidative phosphorylation system and also regulate the expression of numerous other genes that involved in the replication of mtDNA [50,51,52]. Tfam is a transcription factor which acts on the promoter in the D-loop region of mtDNA, and it regulates the replication and transcription of mitochondrial genome [50,52]. *Tfam* gene contains consensus-binding sites for both NRF1 and NRF2, offering an exclusive mechanism for living cells to integrate nuclear DNA-encoded proteins with the transcriptional factor for mtDNA generation [50,52]. A previous study showed that PGC-1α binds to and co-activates the transcriptional function of NRF1 on the promoter for Tfam [53]. In the present study, we demonstrated the activation of mitochondrial biogenesis machinery in the hippocampal CA3 subfield after experimental status epilepticus. Both NRF1 protein expression and NRF1 DNA binding activity increased after KA treatment. In accordance with the NRF1 findings, Tfam, the regulator of the mitochondrial genome, showed similar expression in the hippocampal CA3 subfield. As such, the increased mitochondrial encoded COX1 may indicate the activation of mitochondrial biogenesis. As previously reported [26,27], the long PCR method quantifies intact mtDNA and could be used a reliable indicator for mitochondrial biogenesis. We then showed the change of mtDNA content under experimental status epilepticus. This evidence strengthens the idea that prolonged epilepsy activates the mitochondrial biogenesis machinery in the hippocampal CA3 subfield, and these changes may have a crucial biological role in KA-induced status epilepticus.

In the present study, we demonstrated that exogenous pretreatment of resveratrol in the hippocampus may increase mitochondrial biogenesis machinery expression, including NRF1 protein expression and DNA binding activity, and expression of Tfam. Exogenous resveratrol also augmented the amount of mtDNA as well as mitochondrial DNA encoded COX1 expression, and alleviated apoptotic-like cell death and neuronal damage following experimental status epilepticus. While the role of resveratrol in neuroprotection is complex and still unclear, numerous studies have demonstrated its beneficial effects through its antioxidant, anti-inflammatory, and metal-chelating properties [14,20]. As PGC-1α is a key player in mitochondrial biogenesis in various neurological conditions [46,54] and resveratrol increased PGC-1α expression and mitochondrial biogenesis in the present study, it is tempting to speculate that there is an intimate relationship between resveratrol and mitochondrial biogenesis machinery expression following status epilepticus. Moreover, our previous studies also indicated that PPARγ may protect against mitochondrial damage through upregulation of Bcl-2, an antiapoptotic protein [43]. Therefore, we suggest that the neuroprotective characteristics of resveratrol may be related to attenuating brain tissue damage by ROS and restoring mitochondrial biogenesis and mitochondrial respiratory functions by activation of PGC-1α and PPARγ signaling pathways under the condition of status epilepticus [5,14,43]. The ability to generate more mitochondria may reflect the body’s ability to cope with various acute and chronic neurological insults such as cerebral ischemia, neurodegenerative diseases, and chronic epileptic seizure disorders.

## 4. Materials and Methods 

### 4.1. Animals

Experimental procedures were carried out in compliance with the guidelines for the care and use of experimental animals endorsed by our institutional animal care committee (project identification code 2008061901 was approved by the constituted research ethics committee at Kaohsiung Chang Gung Memorial Hospital, Taiwan; approved on 19 June 2006). Specific pathogen-free adult male Sprague Dawley rats (250–320 g) were purchased from BioLASCO Taiwan Co. Ltd. (Taipei, Taiwan) and housed in an environmentally controlled room (24 ± 1 °C; 12 h/12 h light/dark cycle) in the Center for Laboratory Animals at Kaohsiung Chang Gung Memorial Hospital. Standard laboratory rat chow and tap water were available ad libitum. All efforts were made to reduce the number of animals used and minimize animal suffering during the experiment.

### 4.2. Experimental Status Epilepticus

We used an experimental model of status epilepticus that we established previously [6,7,8]. Briefly, the animal’s head was fixed to a stereotaxic head holder (Kopf, Tujunga, CA, USA) after administering 3% of isoflurane via inhalation to induce anesthesia, and the body was placed on a heating pad to maintain body temperature at 37 °C. Kainic acid (KA; 0.5 nmol; Tocris Cookson, Bristol, UK) dissolved in 0.1 M phosphate buffered saline (PBS, pH 7.4) was microinjected stereotaxically (3.3–3.6 mm posterior to bregma, 2.4–2.7 mm from the midline and 3.4–3.8 mm below the cortical surface) into the CA3 subfield of the hippocampus on the left side. This consistently resulted in progressive and concomitant increases in both root mean square and mean power frequency values of bilateral seizure-like hippocampal electroencephalographic (hEEG) activity recorded from the CA3 subfield on the right side [6,7,8]. According to standard procedure, these experimental manifestations of continuous seizure activity were followed for 60 min, after which they were terminated by intraperitoneal administration of diazepam (30 mg/kg) [6,7,8]. The wound was then closed in layers, and sodium penicillin (10,000 IU; YF Chemical Corporation, Taipei, Taiwan) was given intramuscularly to prevent postoperative infection. The animals were returned to the animal room for recovery in individual cages. Animals that received anesthesia and surgical preparations without additional experimental manipulations served as sham controls.

### 4.3. Pharmacological Pretreatments 

The test agent included a PGC-1α activator, resveratrol (R5010, Sigma-Aldrich, St. Louis, MO, USA) [17,55] that was microinjected bilaterally and sequentially into the CA3 subfield of the hippocampus. The dose of resveratrol used was 100 µmol, at a volume of 150 nL on each side. Microinjection of 3% DMSO (solvent) served as the vehicle and volume control. To avoid the confounding effects of drug interaction, each animal received only one single pharmacological pretreatment, followed 30 min later by microinjection of KA (0.5 nmol) or PBS into the left hippocampal CA3 subfield.

### 4.4. Collection of Tissue Samples from the Hippocampus

At time intervals (1, 3, 6, or 24 h or 7 days) after microinjection of KA or PBS into the hippocampus, rats were anesthetized with 3% isoflurane and perfused intracardially with 50 mL of warm (37 °C) saline containing heparin (100 U/mL). The brain was rapidly removed under visual inspection and placed on a piece of gauze moistened with ice-cold 0.9% saline. We routinely collected tissues from the right hippocampal CA3 subfield (hEEG recording side). This allowed us to ascertain that the results from the analysis were due directly to prolonged seizures and not indirectly to KA toxicity [5,56]. Hippocampal samples were stored at −80 °C prior to use in biochemical analysis.

### 4.5. RNA Isolation and Reverse Transcription Real-Time Polymerase Chain Reaction

For quantitative analysis of *pgc-1α* mRNA expression in the hippocampal CA3 subfield, at 1, 3, 6, or 24 h after microinjection of KA or PBS into the hippocampus, the brain was rapidly removed and total RNA from the hippocampal CA3 was isolated with an RNeasy mini Kit (Qiagen, Dusseldorf, Germany) according to the manufacturer’s protocol as per our previous report [56]. Reverse transcriptase (RT) reaction was performed using an ImProm II^TM^ Reverse Transcription System (Promega, Madison, WI, USA) for first-strand cDNA synthesis. Real-time polymerase chain reaction (PCR) for cDNA amplification was performed using a Roche LightCycler^®^ 480 II system (Roche Ltd., Basel, Switzerland). PCR for sample was performed in duplicate for the cDNAs and glyceraldehyde-3-phosphate dehydrogenase (GAPDH) control [56]. The primer pairs for amplification of *pgc-1α* and GAPDH cDNA used in this study were as follows: *pgc-1α*: forward: 5′-GTTTCATTACCTACCGTTACAC-3′; reverse: 5′-ATCGTCTGAGTTTGAATCTAGG-3′. GAPDH: forward: 5′-CAACTCCCATTCTTCCACCT-3′; reverse: 5′-GCCATATTCATTGTCATACCAG-3′.

For further confirmation of amplification specificity, the PCR products were later subjected to agarose gel electrophoresis [56]. The fluorescence signal from the amplified products was quantitatively measured using the LightCycler software program (Version 3.5, Roche Diagnostics, Mannheim, Germany). Baseline adjustment set in the arithmetic mode was chosen for the second derivative maximum mode. Fold-change analysis was determined the relative change in *pgc-1α* mRNA expression [56], in which fold change = 2^−[ΔΔC*t*]^, where ΔΔC*t* = (C*t*,*_pgc-1α_* − C*t*,_GAPDH_). Note that C*t* value is the cycle number at which the fluorescence signal crosses the threshold.

### 4.6. Western Blot Analysis

Western blot analysis was carried out on proteins extracted from total lysate or from nuclear, mitochondrial, or cytosolic fractions of hippocampal samples. The primary antiserum used included mouse monoclonal or polyclonal antiserum against Tfam (ab131607, Abcam, Cambridge, UK) or COX1 (35-8100, Invitrogen, Grand Island, NY, USA), or rabbit polyclonal antiserum against PGC-1α (sc-13067, Santa Cruz), NRF1 (sc-33771, Santa Cruz Biotechnology, Dallas, TX, USA), nitrotyrosine (A-21285, Invitrogen), or β-actin (ab8227, Abcam). This was followed by incubation with secondary antiserum including horseradish peroxidase (HRP)-conjugated goat anti-mouse IgG (115-035-003, Jackson ImmunoResearch, West Grove, PA, USA) for SIRT1, Tfam, and COX1, or goat anti-rabbit IgG (111-035-045, Jackson ImmunoResearch) for PGC-1α, NRF1, COX IV, and β-actin. Specific antibody–antigen complex was detected by an enhanced chemiluminescence western HRP substrate (Merck Millipore, Burlington, MA, USA). The amount of protein was quantified by ImageJ software (National Institutes of Health, Bethesda, MD, USA), and was expressed as the ratio relative to β-actin protein or mitochondrial control, COX IV.

### 4.7. Double Immunofluorescence Staining and Laser Confocal Microscopy

Double immunofluorescence staining [5,7,8,43,56] was carried out using a goat polyclonal antiserum against PGC-1α and NRF1 (Santa Cruz Biotechnology) and a mouse monoclonal antiserum against a specific neuronal marker, neuron-specific nuclear protein (NeuN; Chemicon). The secondary antisera included goat anti-rabbit IgG conjugated with AlexaFluor 488 and goat anti-mouse IgG conjugated with Alexa Fluor 568 (Molecular Probes, Eugene, OR, USA). Sections were viewed under an Olympus AX-51 epifluorescence microscope (Olympus, Kyoto, Japan); immunoreactivity for NeuN exhibited red fluorescence and PGC-1α manifested green fluorescence.

### 4.8. Electrophoretic Mobility Shift Assay (EMSA)

We measured NRF1 binding activity in nuclear protein from hippocampus following experimental lobe status epilepticus by using electrophoretic mobility shift assay (EMSA) [56]. The following oligonucleotides were employed in binding assays after hybridization to obtain the corresponding DNA duplex: NRF1 cons5, 5′-TCAGAGGGGCCTGCGGCTAT-3′ and NRF1 cons3, 5′-ATAGCCGCAGGCCCCTCTGA-3′.

The oligonucleotides were labeled with Dig-ddUTP solution according to the manufacturer’s recommendation (Roche Molecular Biochemicals, Mannheim, Germany). Binding reactions were performed in a mixture of 20 μL containing binding buffer (10 mM Tris-HCl, 20 mM NaCl, 1 mM DTT, 1 mM EDTA, and 5% glycerol, pH 7.6), 0.5 ng of Dig-labeled probe, 30 μg of nuclear proteins, and 1 μg of poly(dI-dC). After incubation for 20 min at room temperature, the mixture was subjected to gel electrophoresis on a nondenaturing 6% polyacrylamide gel at 180 V for 2 h under a low ionic strength condition. The gel was transferred to positively charged nylon membrane and cross-linked by a UV cross-linker. For competitive binding assay, 100-fold excessive unlabeled oligonucleotides were included for EMSA. For super-shift assays, the binding reaction was conducted with the addition of an antibody specific for the NRF1 (1 μg/reaction) 1 h prior to the introduction of labeled oligonucleotide probes.

### 4.9. Long PCR for Quantitation of Mitochondrial DNA

The long PCR method yielded reliable quantification of virtually completely intact rat mitochondrial DNA (mtDNA) with the use of mouse mtDNA as an internal control, as previously reported [27], with modification. Briefly, the reaction mixtures included 0.4 ng of rat total DNA, 4 pmol of each oligonucleotide primer, 400 mmol/L of dNTP mixture, and 0.5 U of LA Taq enzyme (Takara Bio., Kusatsu, Japan) with a total volume of 10 mL. The same amount (0.4 ng) of total DNA obtained from mouse brains serving as an internal standard was added to the PCR reaction mixture. The primers to amplify 14.3-kb mitochondrial genomes for both rat and mouse were 5′-ATATTTATCACTGCTGAGTCCCGTGG-3′ (forward) and 5′-AATTTCGGTTGGGGTGACCTCGGAG-3′ (reverse). The conditions for long PCR were as previously reported [27] and the PCR products were digested with the restriction enzyme *NcoI* (Promega, Madison, WI, USA) at 37 °C for 2 h and fractionated through 1% agarose gel. While the 14.3 kb fragment was amplified from rat brain mtDNA, the 7.0 and 7.3 kb restrictions were fragmented as one band, representing that the amplified mouse mtDNA served as an internal control [26,27]. The signal intensities of these bands were assessed by image analysis, followed by quantitative densitometry with ImageJ (National Institutes of Health, Bethesda, MD, USA).

### 4.10. Qualitative and Quantitative Analysis of DNA Fragmentation

After extraction of total DNA from hippocampal tissues, nucleosomal DNA ladders were amplified by a PCR kit for DNA ladder assays (Maxim Biotech, San Francisco, CA, USA) to enhance the detection sensitivity, and were separated by electrophoresis on 1% agarose gels [7,56]. To quantify apoptosis-related DNA fragmentation, a cell death ELISA (Roche Molecular Biochemicals, Mannheim, Germany) was used to assay the level of histone-associated DNA fragments in the cytoplasm. Proteins from hippocampal samples were used as the antigen source, together with primary anti-histone antibody and secondary anti-DNA antibody coupled to peroxidase. The amount of nucleosomes in the cytoplasm was quantitatively determined using 2,2’-azino-di-[3-ethylbenzthiazoline] sulfonate as the substrate. Absorbance was measured at 405 nm and referenced at 490 nm using a Multiskan Spectrum reader (Thermo Scientific, Miami, OK, USA).

### 4.11. Immunofluorescent Staining Analysis of Apoptotic Neuronal Cells

The removed brain tissue was fixed in 4% formaldehyde for 18 h at 4 °C and cryoprotected in 30% sucrose solution in PBS. Frozen transverse sections (30 μm) at the level of the hippocampus were cut on a cryostat and collected in 0.1 M PBS. Free-floating sections of the hippocampus were processed with an in situ cell death detection kit for immunoreactivity for TUNEL (TMR red, Roche-12156792910, Sigma-Aldrich, St. Louis, MO, USA); sections were also stained with DAPI (Sigma-Aldrich). Sections were viewed under an Olympus AX-51 epifluorescence microscope (Olympus, Kyoto, Japan).

### 4.12. Statistical Analysis

The continuous variables were expressed as mean ± standard error of the mean (SEM). One-way analysis of variance followed by Scheffé multiple range tests for post hoc assessment of individual means were used to compare the group mean differences. *p* < 0.05 was considered statistically significant.

## 5. Conclusions

Resveratrol plays a pivotal role in mitochondrial biogenesis, which may present a neuroprotective mechanism counteracting seizure-induced neuronal damage by activation of the PGC-1α signaling pathway. The ability to enhance resveratrol-PGC-1α signaling may uncover important potential in managing patients with status epilepticus.

## Figures and Tables

**Figure 1 ijms-20-00998-f001:**
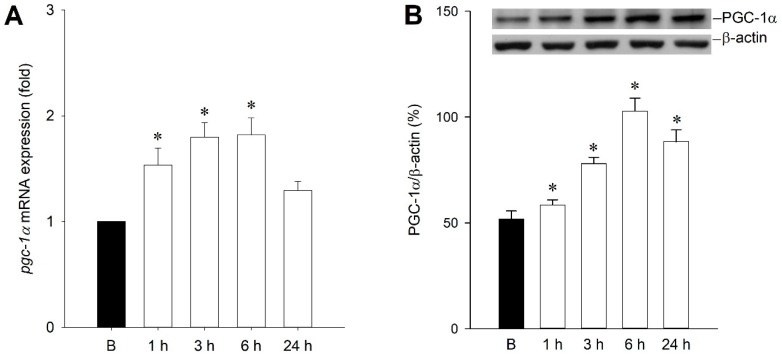
(**A**) Upregulation of expression of *pgc-1α* mRNA, and (**B**) changes in PGC-1α protein relative to β-actin after microinjection of kainic acid (KA) in hippocampal CA3 subfield. Samples were collected from the right CA3 subfield of the hippocampus at 1, 3, 6, or 24 h after microinjection of 0.5 nmol KA or phosphate buffered saline (PBS) into the left hippocampal CA3 subfield. Values are mean ± standard error of the mean (SEM) of quadruplicate analyses from six animals per experimental group. * *p* < 0.05 versus sham-control group in the Scheffé multiple-range test.

**Figure 2 ijms-20-00998-f002:**
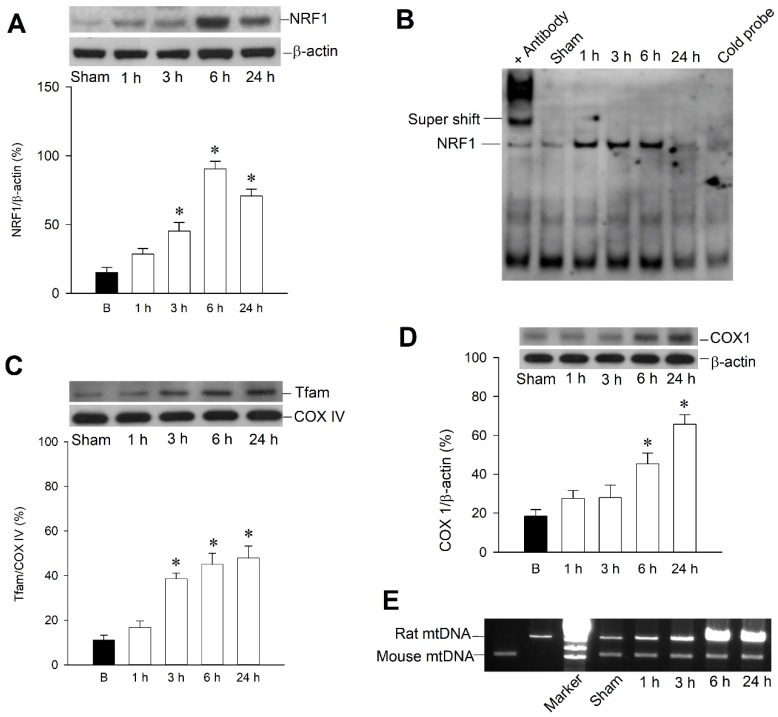
Involvement of mitochondrial biogenesis in kainic acid (KA)-induced status epilepticus in hippocampal CA3 subfield. (**A**) Temporal changes in nuclear respiratory factor 1 (NRF1) protein relative to β-actin protein. (**B**) Representative gel depicting electrophoresis mobility shift assay of NRF1 DNA binding activity in nuclear extracts from right CA3 subfield of hippocampus 1–24 h after microinjection of KA (0.5 nmol) into left hippocampal CA3 subfield. (**C**) Mitochondrial fraction of samples collected 1–24 h after microinjection of KA (0.5 nmol) or PBS into left hippocampal CA3 subfield for mitochondrial transcription factor A (Tfam) expression. Cytochrome c oxidase IV (COX IV) was used as internal loading control for mitochondrial fraction. (**D**) Temporal changes in COX I protein relative to β-actin protein. (**E**) Long PCR for quantitation of mitochondrial DNA revealed temporal change after microinjection of KA (0.5 nmol) or PBS into left hippocampal CA3 subfield. Values are mean ± SEM of the ratio of β-actin or COX IV to loading controls and are quadruplicate analyses from six animals per experimental group in (**A**,**C**,**D**). * *p* < 0.05 versus sham-control group in the Scheffé multiple-range test.

**Figure 3 ijms-20-00998-f003:**
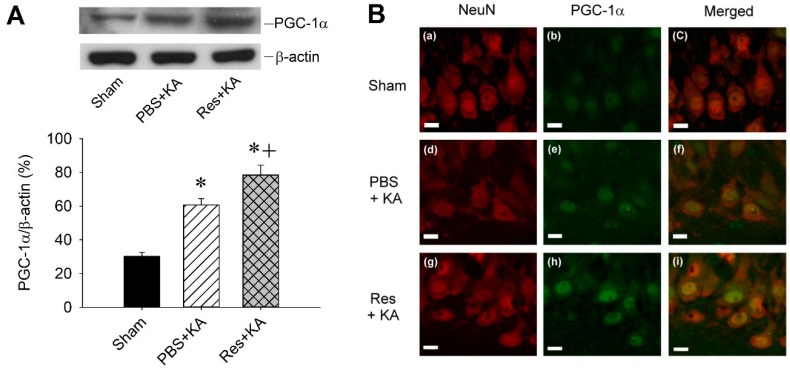
(**A**) Representative gels (inset) or changes in PGC-1α relative to β-actin from CA3 subfield of hippocampus 6 h after microinjection of 0.5 nmol kainic acid (KA) or pretreatment with microinjection of PGC-1α activator, resveratrol (Res; 100 μmol) into hippocampal CA3 subfield. Values are mean ± SEM of quadruplicate analyses from six animals per experimental group. * *p* < 0.05 versus sham control group, + *p* < 0.05 versus PBS + KA group in the Scheffé multiple-range test. (**B**) Laser scanning confocal microscopic images of right CA3b subregion of hippocampus showing cells immunoreactive to a neuronal marker, NeuN (red fluorescence), or additionally stained for PGC-1α (green fluorescence) from sham animals (a–c), or 6 h after microinjection of 0.5 nmol KA (d–f) or pretreatment with microinjection of resveratrol (Res; 100 µmol) before KA into the hippocampal CA3 subfield (g–i). Scale bar, 10 μm.

**Figure 4 ijms-20-00998-f004:**
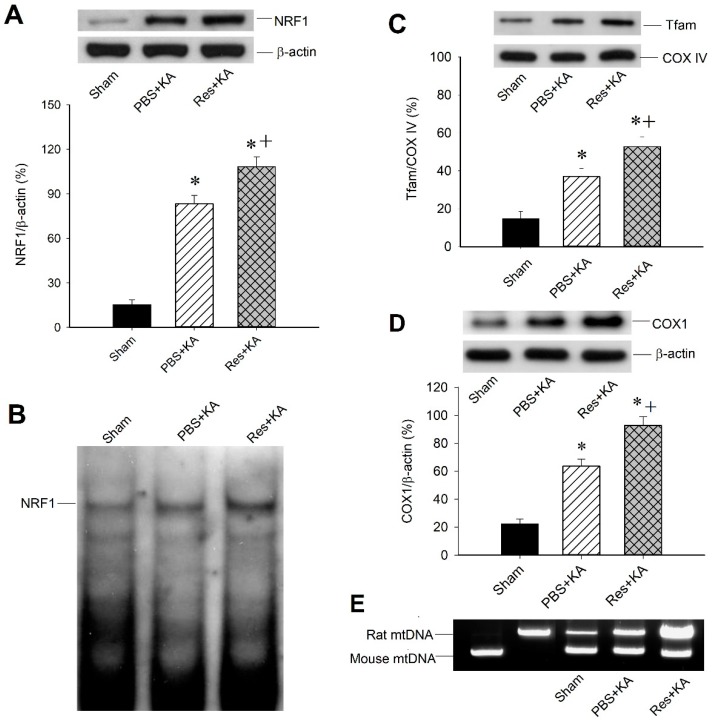
Effects of resveratrol on mitochondrial biogenesis in kainic acid (KA)-induced status epilepticus in hippocampal CA3 subfield. (**A**) Pretreated microinjection with resveratrol (Res, 100 µmol) into the hippocampus increased KA-induced NRF1 expression. (**B**) In accordance with NRF1 protein expression, DNA binding activity of NRF1 measured by electrophoretic mobility shift assay also increased with microinjection with resveratrol (100 µmol) into the hippocampus. (**C**) Pretreated microinjection with resveratrol (100 µmol) into the hippocampus enhanced Tfam expression 24 h after KA treatment. (**D**) Mitochondrial protein, COX1, and (**E**) mtDNA content increased in rats with pretreated microinjection with resveratrol (100 µmol) into the hippocampus. Values are mean ± SEM of quadruplicate analyses from six animals per experimental group. * *p* < 0.05 versus sham control group, + *p* < 0.05 versus PBS + KA group in the Scheffé multiple-range test.

**Figure 5 ijms-20-00998-f005:**
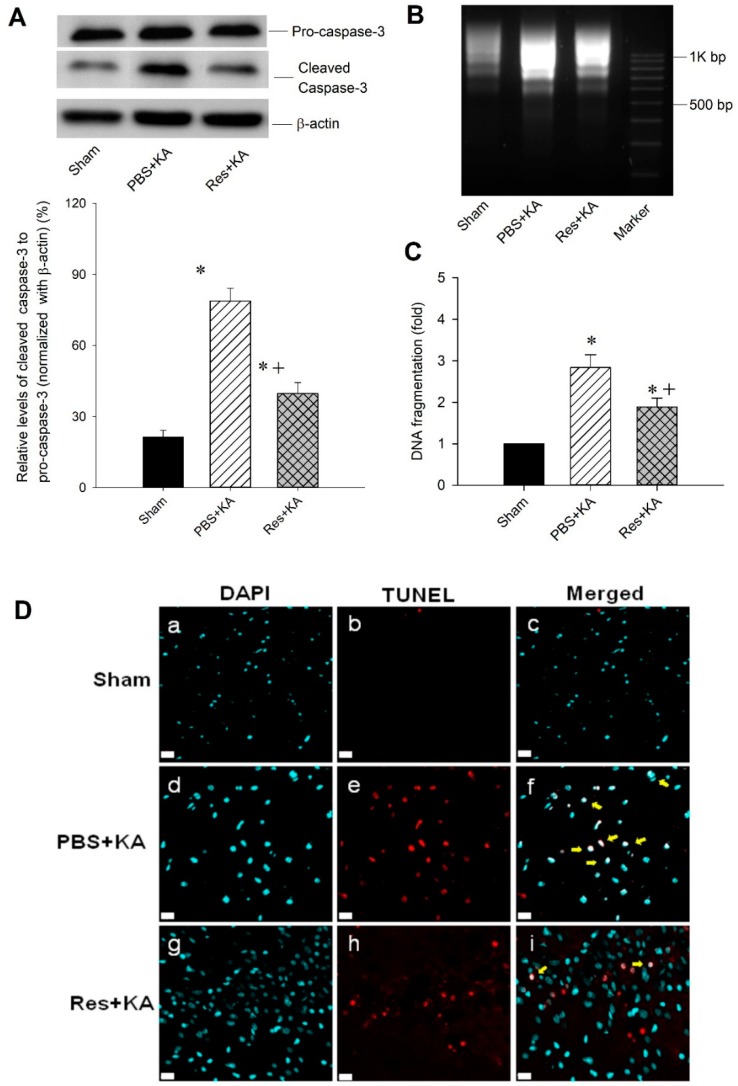
Resveratrol reduced seizure-related neuronal damage in in kainic acid (KA)-induced status epilepticus in hippocampal CA3 subfield. (**A**) Representative gels (inset) or changes in activated caspase-3 to pro-caspase-3 (normalized with β-actin) detected in the cytosolic fraction of samples collected from the CA3 subfield of hippocampus in animals with sham-controls 7 days after microinjection of KA (0.5 nmol) into the left hippocampal CA3 subfield with PBS or resveratrol (Res, 100 µmol) pretreatment. (**B**) Qualitative and (**C**) quantitative analysis of DNA fragmentation detected in samples collected from the CA3 subfield of hippocampus 7 days after induced status epilepticus. Values are mean ± SEM of quadruplicate analyses from six animals per experimental group. * *p* < 0.05 versus sham control group, + *p* < 0.05 versus PBS + KA in the Scheffé multiple-range test. (**D**) Immunofluorescent staining showed terminal deoxynucleotidyl transferase dUTP nick end labeling (TUNEL) immunoreactivity in hippocampal CA3b neurons on the right side, by co-immunofluorescence staining with 4′,6-diamidino-2-phenylindole (DAPI) (arrows), in sham-control animals (a–c), or 7 days after microinjection of KA in animals that received pretreatment with PBS (d–f) or with resveratrol (Res, 100 µmol) (g–i). Scale bar, 10 μm.

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
