# Peer review of "Resveratrol Promotes Mitochondrial Biogenesis and Protects against Seizure-Induced Neuronal Cell Damage in the Hippocampus Following Status Epilepticus by Activation of the PGC-1α Signaling Pathway"

_ijms, 2019, doi:10.3390/ijms20040998_

Round 1
Reviewer 1 Report
comment:
please include in the introduction the paragraph summarising information about polyphenols (what are, classes, effect of human health etc.) for the description of polyphenols: [PMID: 22794138], for effect on human health: [PMID: 23953879; PMID: 28472215; PMID: 29695122; PMID: 26576219]
Author Response
Responses to Editor:
The details of approval by a properly constituted research ethics have listed in P. 13, Lines 377-379.
Responses to Reviewer #1:
1. please include in the introduction the paragraph summarising information about polyphenols (what are, classes, effect of human health etc.) for the description of polyphenols: [PMID: 22794138], for effect on human health: [PMID: 23953879; PMID: 28472215; PMID: 29695122; PMID: 26576219]
Response: we have included the paragraph to summarize the information about polyphenols and added these references (P. 2, Lines 70-81).

Reviewer 2 Report
Dear Authors, please, check on these issues:
Abstract: "Peroxisome proliferator-activated receptor gamma coactivator-1α (PGC-1α)...".
Introduction: lines 55-56 must be rewritten.
Line 70: "...resveratrol can activate sirtuin 1 (SIRT1), which is a class III lysine-deacetylase [15]".
Line 95: "...protein levels in total proteins extracted from the right hippocampal...".
Line 107: please, write in extenso NRF1 the first time.
Figure 2A: it refers to NRF1, not SIRT1 as written in the text. Furthermore, the sentence in line 109 reports a peak after 8h. However, Fig. 2A does not show this time point.
Line 114: Tfam must be written in extenso the first time.
Lines from 113-124: the message is confused. I recommend an English proofreading.
Line 135: "Effect of resveratrol on PGC-1α expression ...".
Figure 4C: it does not seem that there is an increase in double immunofluorescence after resveratrol pre-treatment. It is difficult to observe differences between the three panels c-f-i.
Figure 5: the differences reported in figure 5A and 5B are very little. For instance, the % increase of ATP after the pre-injection with resveratrol looks 02-0.3%. These are not really increases. Figure 5C is also not clear. What is the "control" used to calculate the ratio in the lower panel?
Figure 6: after how many days the evaluation of caspase-3 was carried out? Also, it should be useful to use an antibody able to detect both pro-caspase-3 and cleaved caspase-3. Additionally, a further apoptosis assay should be performed. I suggest a flow cytometry assay by using AnnV /P.I. Otherwise, you could test your samples by nuclear staining with DAPI or Hoechst in order to assess the changes in apoptotic bodies. In any case, the information concerning apoptosis should be confirmed and improved.
Author Response
Responses to Editor:
The details of approval by a properly constituted research ethics have listed in P. 13, Lines 377-379.
Responses to Reviewer #2:
1. Abstract: "Peroxisome proliferator-activated receptor gamma coactivator-1α (PGC-1α)...".
Response: we have corrected this mistake (Line 29).
2. Introduction: lines 55-56 must be rewritten.
Response: we have rewritten this sentence (Line 60-64)
3. Line 70: "...resveratrol can activate sirtuin 1 (SIRT1), which is a class III lysine-deacetylase [15]".
Response: we have corrected this mistake (Line 86).
4. Line 95: "...protein levels in total proteins extracted from the right hippocampal...".
Response: we have corrected this mistake (Line 110).
5. Line 107: please, write in extenso NRF1 the first time.
Response: we have corrected and indicated (Line 122).
6. Figure 2A: it refers to NRF1, not SIRT1 as written in the text. Furthermore, the sentence in line 109 reports a peak after 8h. However, Fig. 2A does not show this time point.
Response: we have corrected this mistake, NRF1, not SIRT1 (Line 124). The time of peaking level is 6 h after KA treatment, we also corrected (Line 124).
7. Line 114: Tfam must be written in extenso the first time.
Response: we have corrected and indicated (Line 129).
8. Lines from 113-124: the message is confused. I recommend an English proofreading.
Response: we will refer this manuscript to “English editing service of the journal” before submission of our revision.
9. Line 135: "Effect of resveratrol on PGC-1α expression ...".
Response: we have corrected this mistake (Line 151).
10. Figure 4C: it does not seem that there is an increase in double immunofluorescence after resveratrol pre-treatment. It is difficult to observe differences between the three panels c-f-i.
Response: Thank you for your suggestion. We have changed the Figure 4B to show the differences between these three groups. Besides, we calculated the mean fluorescence intensity of NRF1 (Figure 4C) to show the differences between these three groups Please see the Lines 186-191; Figure 4 B and C.
11. Figure 5: the differences reported in figure 5A and 5B are very little. For instance, the % increase of ATP after the pre-injection with resveratrol looks 02-0.3%. These are not really increases. Figure 5C is also not clear. What is the "control" used to calculate the ratio in the lower panel?
Response: Thank you for your comments. We may provide wrong version of plot in Figure 5B. According to our data, the decrease of about 50% of ATP levels was found in animals following 3 days after KA-induced status epilepticus compared with sham control animals. Otherwise, pretreatment with microinjection of resveratrol (100 mmol) into bilateral hippocampus also recovery to nearly 70% of normal ATP concentrations in damaged hippocampal tissues that induced by experimental status epilepticus. In our original plot, following the title of y-axis, the scale of y-axis is percentage, however, the numbers of y-axis apparently indicated the ratio to the sham group. The incorrect combination made wrong information we tried to conduct. To revise this misleading combination, we replaced the correct numbers to the y-axis (Lines 221-234). For Figure 5C, we chose sham control to be our control group throughout this study and initiated to “Sham” in this article. In Figure 5C, all the values quantified with blots were subtracted with sham control group to acquire the ratio. For the consistency, we replaced the title of y-axis in 5C.
12. Figure 6: after how many days the evaluation of caspase-3 was carried out? Also, it should be useful to use an antibody able to detect both pro-caspase-3 and cleaved caspase-3. Additionally, a further apoptosis assay should be performed. I suggest a flow cytometry assay by using AnnV /P.I. Otherwise, you could test your samples by nuclear staining with DAPI or Hoechst in order to assess the changes in apoptotic bodies. In any case, the information concerning apoptosis should be confirmed and improved.
Response: Thank you for your comments and suggestion. All of the analysis of cell death was measured 7 days after KA-induced experimental status epilepticus. We have indicated the information in the results (Lines 254, 257, 268-9 and 271-2), and Figure 6. We also added the further information of apoptosis assay by immunofluorescent staining showed terminal deoxynucleotidyl transferase dUTP nick end labeling (TUNEL) immunoreactivity in hippocampal CA3b neurons and co-immunofluorescence staining with 4’,6-diamidino-2-phenylindole (DAPI) (Figure 6D; Result: Lines 259-263; and Methods: 541-548).

Round 2
Reviewer 2 Report
Dear Authors,
according to the cover letter and revised version that you provided, I recommend that:
- Figure 4: I still do not see differences in the fluorescence intensity of panel 4B (between KA or Res+KA). If these are the best images you have, I would rather cut this information.
I agree with all the western blots of Figure 4 that show consistent increases of NRF1, Tfam and COX1.
-Figure 5: the overall data are not convicing to me. Especially the oxidized protein levels do not seem to be convincing. There are not really clear changes (Fig. 5C). I would not report these data.
Author Response
Responses to Reviewer #2:
Response: We have referred this manuscript to MDPI English Editing Service and Language has been corrected (English editing ID: English-8029).
1. Figure 4: I still do not see differences in the fluorescence intensity of panel 4B (between KA or Res+KA). If these are the best images you have, I would rather cut this information.
Response: Thank you for your comment. As your suggestion, we have cut the information of immunohistochemistry figure in Figure 4 to avoid the confounding (Lines 178-191; 193-202).
2. Figure 5: the overall data are not convicing to me. Especially the oxidized protein levels do not seem to be convincing. There are not really clear changes (Fig. 5C). I would not report these data.
Response: Thank you for your comments. In the previous studies, we have been showed depressed the Complex I activity and increased oxidized protein level following status epilepticus. In this study, resveratrol showed partially improved the mitochondrial functions and oxidative stress. Since these data cannot be convincing, and they are additional data, we have deleted these results in Figure 5 and also revised the results, discussion, methods and abstract.
We thus appreciate very much the opportunity to improve on our manuscript; and sincerely hope that our revision will now meet with your approval for publication in International Journal of Molecular Sciences.

Round 3
Reviewer 2 Report
Dear Authors,
I read the changes you made to your manuscript.